# Extending methane profiles from aircraft into the stratosphere for satellite total column validation using the ECMWF C-IFS and TOMCAT/SLIMCAT 3D model

Shreeya Verma [1], Julia Marshall [1], Mark Parrington[2], Anna Agustí-Panareda[2], Sebastien Massart[2], Martyn P. Chipperfield[3], Christopher Wilson[3], Christoph Gerbig[1]

[1] Max Planck Institute for Biogeochemistry, Jena, Germany.
[2] European Centre for Medium-Range Weather Forecasts, Reading, UK.
[3] National Centre for Earth Observation, School of Earth and Environment, University of Leeds, UK.

*Correspondence to*: S. Verma (sverma@bgc-jena.mpg.de)

**Abstract.** Airborne observations of greenhouse gases are a very useful reference for validation of satellite-based column-averaged dry air mole fraction data. However, since the aircraft data are available only up to about 9-13 km altitude, these profiles do not fully represent the depth of the atmosphere observed by satellites and therefore need to be extended synthetically into the stratosphere. In the near future, observations of $CO_2$ and $CH_4$ made from passenger aircraft are expected to be available through the In-Service Aircraft for a Global Observing System (IAGOS) project. In this study, we analyse three different data sources that are available for the stratospheric extension of aircraft profiles by comparing the error introduced by each of them into the total column and provide recommendations regarding the best approach. First, we analyse $CH_4$ fields from two different models of atmospheric composition - the European Centre for Medium-Range Weather Forecasts (ECMWF) Integrated Forecasting System for Composition (C-IFS) and the TOMCAT/SLIMCAT 3-D chemical transport model. Secondly, we consider scenarios that simulate the effect of using $CH_4$ climatologies such as those based on balloons or satellite limb soundings. Thirdly, we assess the impact of using a-priori profiles used in the satellite retrievals for the stratospheric part of the total column. We find that the models considered in this study have a better estimation of the stratospheric $CH_4$ as compared to the climatology-based data and the satellite a-priori profiles. Both the C-IFS and TOMCAT models have a bias of about -9 ppb at the locations where tropospheric vertical profiles will be measured by IAGOS. The C-IFS model, however, has a lower random error (6.5 ppb) than TOMCAT (12.8 ppb). These values are well within the minimum desired accuracy and precision of satellite total column $XCH_4$ retrievals (10 ppb and 34 ppb, respectively). In comparison, the a-priori profile from the University of Leicester Greenhouse Gases Observing Satellite (GOSAT) Proxy $XCH_4$ retrieval and climatology-based data introduce larger random errors in the total column, being limited in spatial coverage and temporal variability. Furthermore, we find that the bias in the models varies with latitude and season. Therefore, applying appropriate bias correction to the model fields before using them for profile extension is expected to further decrease the error contributed by the stratospheric part of the profile to the total column.

## 1 Introduction

Space-based observations of atmospheric greenhouse gases hold great potential for gaining a better understanding of the dynamics of the global carbon cycle. Satellite measurements such as those from the Greenhouse Gases Observing Satellite (GOSAT) and the Orbiting Carbon Observatory-2 (OCO-2) provide column-averaged dry air mole fractions of $CO_2$ ($XCO_2$) and $CH_4$ ($XCH_4$) (Yokota et al., 2009, Yoshida et al., 2011) that can be used in inverse simulations to estimate carbon sources and sinks at the Earth's surface along with their spatial and temporal distributions.

A precondition for the use of satellite-based total column observations in inverse modelling studies is that these measurements must be sufficiently accurate and precise. Rayner and O'Brien (2001) have shown that the precision

requirement for remotely sensed total column-integrated $CO_2$ abundances to be useful in constraining surface fluxes is less than 1 % (3-4 ppm), while others (e.g. Miller et al., 2007) suggest even more stringent requirements (1-2 ppm). For total column abundance of $CH_4$, the required precision of these measurements is around 34 ppb or less (Buchwitz et al., 2011). Hence, before these space-based observations can be used for flux estimation, they must be validated and calibrated using
independently obtained measurements of even higher precision.

To this end, in-situ measurements made by sensors deployed on aircraft have proved to be extremely useful. These measurements are currently being used in addition to ground-based remote sensing total column data such as those from the Total Carbon Column Observing Network (TCCON), a network of ground-based Fourier Transform Spectrometers that provides valuable reference data for validation of satellite total column retrieval, currently at 23 sites across the globe
(Wunch et al., 2011). However, these data further depend on in-situ measurements made from aircraft or AirCore (Karion et al., 2010) for validation and calibration (Wunch et al., 2010; Geibel et al., 2012).

There have been a number of recent studies that have used airborne measurements from commercial aircraft and research aircraft campaigns. Inoue et al., (2016) used TCCON measurements for bias correcting total column $XCH_4$ and $XCO_2$ retrievals from GOSAT and further verified the approach using aircraft measurements. Inoue et al., (2013) and Miyamoto et
al., (2013) were focused on validation of GOSAT $XCO_2$ while de Laat et al., (2012) and de Laat et al., (2014) presented a validation approach using commercial aircraft profiles for CO measurements from SCIAMACHY and MOPITT respectively. While both commercial aircraft and research aircraft provide accurate, high-resolution in-situ atmospheric information, operational commercial aircraft measurements have the added advantage of global coverage and availability over long periods of time (Petzold et al., 2015). The In-Service Aircraft for a Global Observing System (IAGOS) project is
a recently established European Research Infrastructure conducting long-term observations of atmospheric species with the help of sensors deployed on board commercial aircraft. While currently it provides for the measurement of species like carbon monoxide (CO), ozone ($O_3$), water vapour ($H_2O$), nitrogen oxides ($NO_x$, $NO_y$) and aerosols, measurements of carbon dioxide ($CO_2$) and methane ($CH_4$) is also foreseen in the near future.

One of the limitations of aircraft profiles as a source of reference data for validation of total column data is that their
altitudinal extent does not represent the full depth of the atmosphere observed by the satellites. The profiles generally do not extend much above the tropopause and have to be extended further into the stratosphere using other sources of information in order to compute the total column abundance. These sources could include model output (de Laat et al., 2012), climatologies based on balloon-borne measurements that measure above the tropopause up to about 30km altitude (Geibel et al., 2012), satellite limb soundings (Inoue et al., 2014) or the stratospheric portion of the a-priori profile used in the satellite
retrieval. Therefore, in order to be able to use the aircraft profiles for validation of satellite columns, we need to choose an appropriate data source for profile extension based on a sound evaluation of the available options and the uncertainty that each of them introduces to the total column.

In this context, $CH_4$ poses more challenges than some other tracers like CO and $CO_2$. $CH_4$ is a critical driver of stratospheric chemistry and is known to have a stratospheric sink due to oxidation reactions with OH (hydroxyl) and Cl (chlorine)
radicals. This fact makes the choice of the stratospheric extension extremely crucial for $CH_4$ when using aircraft profiles for validation of total column observations. This is because, although the stratosphere has a small mass relative to the total column, chemical losses in the stratosphere result in a steep gradient in the $CH_4$ mixing ratio with height. Misrepresentation of this gradient in the stratospheric extent can have a major impact on the calculated column-integrated concentration. Wunch et al., (2010) showed that the contribution of the error from the unsampled part of the atmosphere above the highest

altitude of the aircraft profiles is the largest towards the error in the total column. Therefore, we need to reasonably estimate and, if possible, reduce the error associated with the stratospheric extension of the aircraft profile. In order to do that a good understanding of the stratospheric dynamics and variability is critical.

So far an analysis of the impact of using different extensions has not been performed and most validation studies using aircraft profiles have used only one data source for the extension of the aircraft column. In this study we evaluate three different potential candidates that can be used as stratospheric extensions for $CH_4$ by quantifying and characterizing the error associated with each. These are: model output, climatologies based on balloon or satellite limb soundings and a-priori profiles from satellite retrievals. The main idea is to quantify the contribution of the bias and variability in the stratospheric column from each of these data sources on the total column abundance of $CH_4$ and, on the basis of this analysis, provide recommendations regarding which of the data sources to use. We also examine regional differences in the applicability of the approach, and identify regions that prove particularly difficult. The uncertainty from each of these data sources is computed using reference data from satellite limb measurements from the Michelson Interferometer for Passive Atmospheric Sounding (MIPAS) (Fischer et al., 2008; Raspollini et al., 2006) instrument which was in operation between 2000 and 2012 and formed a part of the core payload of ENVISAT (Environmental Satellite). In order to get realistic estimates and distribution of the stratospheric uncertainty introduced in $XCH_4$, we estimate the magnitude of the error associated with each data source at real aircraft profile locations coming from the Measurement of OZone and water vapour by AIrbus in-service airCraft (MOZAIC) project (Marenco et al., 1998). The project started in 1993 with the aim of collecting $O_3$, $H_2O$, CO and $NO_y$ data with the help of high tech sensors deployed onboard five long-range commercial airliners. This project is the predecessor of the IAGOS project and hence the sampling is expected to be comparable to that from IAGOS.

The model output analysed in this study is obtained from two models:

1. The Integrated Forecasting system for Composition (C-IFS) (Flemming et al., 2015; Massart et al. 2014) is a comprehensive, state of the art numerical weather prediction (NWP) and Earth-system model developed at the European Centre for Medium - Range Weather Forecasts (ECMWF). It models the dynamics of the atmosphere and the physical processes that influence the weather as well as the atmospheric composition. Data assimilation of meteorological and atmospheric $CH_4$ observations from the SRON product of GOSAT (Butz et al., 2010) is used in order to produce a global atmospheric $CH_4$ analysis based on an optimal estimation of the state of the atmosphere.

2. The TOMCAT/SLIMCAT model (Chipperfield, 1999; 2006), a 3-D offline chemistry transport model that simulates the temporal and spatial distribution of chemical tracers in the troposphere and stratosphere. The model has a detailed chemistry scheme and is driven by winds and temperature fields obtained from the ERA-Interim meteorological reanalysis.

As a sanity check, we also compare the model bias to that obtained using $CH_4$ profiles from the ACE-FTS instrument (Bernath et al., 2005) on the Canadian satellite SCISAT-1, launched in August 2003 with the main goal of studying the chemical and dynamical processes that impact stratospheric ozone depletion.

Since climatology-based data are long-term averages, generally with sparse spatial coverage, we investigate the impact of using these data for the stratosphere by simulating the effect of temporal averaging and reduced spatial coverage on the stratospheric column error. For this, we analyse the error introduced by the following: 1) Monthly mean $CH_4$ fields from the C-IFS model. 2) Monthly mean C-IFS fields based on sampling as that of the (a) ACE-FTS and (b) MIPAS instruments for the stratosphere. This helps to quantify how much uncertainty is introduced if there is a poorer representation of the $CH_4$ variability in the data and if the spatial coverage of the data is low. Further, it allows us to determine if it is better to use the

full variability of $CH_4$ from a (potentially biased) model rather than the lower-bias monthly means lacking temporal variability from mean satellite fields. It is noteworthy that the idea behind option 2) is to not compare the impact of using the profiles from the two instruments per se, since MIPAS is no longer flying and hence cannot be used for profile extension in the future, but to evaluate the effect of the different type of sampling from the two instruments i.e. ACE-FTS-like (sparse) and MIPAS-like (dense). Since there is no realistic "truth" of MIPAS or ACE measurements at all times and all places throughout the month, here the full C-IFS fields are treated as the truth and compared to monthly mean fields derived from the C-IFS sampled at the MIPAS and ACE-FTS locations and times. Thus, for this part of the study, no actual climatology data are used and only the uncertainty introduced by the sampling and averaging is assessed. The computed error in the two cases is then re-calculated with respect to MIPAS using the bias in the full C-IFS fields obtained from comparison with MIPAS.

Lastly, the stratospheric column uncertainty from using the a-priori profile of the satellite retrieval for profile extension is estimated. This is achieved using the University of Leicester GOSAT Proxy $XCH_4$ retrieval (Parker et al., 2011).

The layout of the paper is as follows. Section 2 describes the different datasets used in the study as well as the methodology and approach. Section 3 presents the details of the stratospheric error estimation and comparison of the different profile extensions. Section 4 presents the discussion and conclusions of our results

## 2 Materials and Methods

### 2.1 Datasets

**Integrated Forecasting system for Composition (C-IFS)**

The Integrated Forecasting System for Composition (C-IFS) is a comprehensive NWP Earth system model developed at the ECMWF. It uses 4D-Var (Rabier et al. 2000) to assimilate data from a wide range of different observation networks and satellite instruments into the model in order to produce optimal estimates of the state of the atmosphere. In addition to this, monitoring of atmospheric composition and modelling of greenhouse gases has also been incorporated into the IFS (Flemming et al., 2015; Massart et al., 2014) as a part of the Copernicus Atmosphere Monitoring Service (CAMS, https://atmosphere.copernicus.eu) and previously the Monitoring of Atmospheric Composition and Climate (MACC, mac.copernicus-atmosphere.eu) projects.

The C-IFS model uses surface $CH_4$ fluxes and loss rate prescribed from inventories and climatologies. The $CH_4$ fluxes are those used as priors for flux estimation in the study by Bergamaschi et al., (2009), except for anthropogenic fluxes which are obtained from the EDGAR 4.2 database (Janssens-Maenhout et al., 2012) for the year 2008, and biomass burning emissions which are taken from the CAMS GFAS data set (Kaiser et al., 2012). For the chemical sink in the troposphere and the stratosphere, the climatological chemical loss rates from Bergamaschi et al., (2009) are used. These are based on OH fields optimised with methyl chloroform using the TM5 model (Krol et al., 2005) and prescribed concentrations of the stratospheric radicals using the 2-D photochemical Max Planck Institute model.

In this study, we diagnose the tropopause height using the humidity gradient from the C-IFS model. The tropopause height is used to separate the tropospheric and stratospheric partial columns of $CH_4$. We use $CH_4$ analysis product from C-IFS that includes the assimilation of the GOSAT $CH_4$ product from SRON (Butz et al., 2010). The model run has a horizontal

Gaussian grid with a resolution of TL255 (~80 km), but the output are averaged onto a regular $1° \times 1°$ grid. The model has 60 vertical levels from the surface up to 0.1 hPa. Temporal resolution of the $CH_4$ analysis fields is 6 hours. The meteorological reanalysis products are used as input for a number of offline transport models and since it provides data at a high vertical and horizontal resolution, it has also been used as a reference for the development of some CTMs, e.g.

TOMCAT/SLIMCAT (described below) and TM5 (Krol et al., 2005).

**TOMCAT/SLIMCAT model**

TOMCAT/SLIMCAT is a three-dimensional off-line chemical transport model (CTM) first described by Chipperfield et al., (1993). The model is driven using prescribed winds and temperatures and simulates the abundances of chemical and aerosol

tracers in the troposphere and stratosphere. The TOMCAT model has been used extensively for chemistry and transport studies in the stratosphere and troposphere (e.g., Stockwell et al., 1999; Monks et al., 2012; Richards et al., 2013; Chipperfield et al., 2015). The TOMCAT version, as used here, employs a hybrid σ-p vertical coordinate system. Tracer advection is based on a conservation of second-order moments scheme described in Prather (1986) and convective transport is based on the mass flux scheme of Tiedtke (1989). In general the model has a flexible vertical and horizontal resolution.

The SLIMCAT model was developed later as the 'stratosphere only' version of the TOMCAT model using a hybrid σ-θ vertical coordinate system. The SLIMCAT model was further developed and extended downwards to include the tropospheric levels to form the unified TOMCAT/SLIMCAT model (Chipperfield, 2006) allowing a choice of the vertical coordinate system.

In this study, output has been taken from a TOMCAT simulation with the moderate horizontal resolution of $2.8° \times 2.8°$ with

32 vertical levels from the surface to 0.1 hPa. The model has a detailed interactive stratospheric chemistry scheme with explicit simulation of the $CH_4$ loss reactions. The model run started in 1979 and was forced by 6-hourly ECMWF ERA-Interim reanalyses. The tropospheric mixing ratios of long-lived source gases, including $CH_4$, $N_2O$ and halocarbons, were specified from monthly global mean observations. The temporal resolution of the available gridded model output is 6 hours.

In the subsequent sections of this paper we will refer to the TOMCAT/SLIMCAT model as 'TOMCAT'. The results of the

TOMCAT simulation are complementary to those from the C-IFS model in the sense that they are obtained from a computationally inexpensive forward CTM, which has no additional constraint such as chemical data assimilation in the stratosphere.

**MIPAS observations of $CH_4$**

MIPAS is a Fourier transform infrared limb emission spectrometer on the ENVISAT (Environmental Satellite) that was operational between 2002 and 2012 (Fischer et al., 2008; Raspollini et al., 2006). It provided trace gas information of a number of species mainly in the upper tropospheric, stratospheric, and mesospheric levels measuring continuously and providing nearly global coverage in a single day. From 2002 to 2004 MIPAS operated at a high spectral resolution mode

(Glatthor et al., 2005), while from 2005 to 2012 its operation was based on the reduced spectral resolution (Chauhan et al., 2009; von Clarmann et al., 2009b)

In this study we use $CH_4$ profiles for the year 2010, from the V5R_CH4_224 version retrieved with the IMK/IAA (Institut für Meteorologie und Klimaforschung, Karlsruhe/Instituto de Astrofisica de Andalucia, Granada) MIPAS scientific level 2 processor. The retrieval algorithm is described in detail in Pleininger et al., (2015). These $CH_4$ profiles are validated in

Pleininger et al., (2016). Although data is provided at a grid that extends from 0 to 120 km, the range over which the data can be considered reliable is only between 13 and 50 km. In order to use the profiles as reference truth for comparison with the $CH_4$ profiles from the C-IFS and TOMCAT models, they are interpolated to the model grid before comparison.

**ACE-FTS observations of $CH_4$**

The ACE-FTS is a limb-sounding instrument on the SCISAT-1 satellite that was launched in August 2003 (Bernath et al., 2005). The satellite operates on a high inclination (74°), circular low Earth orbit. The ACE – FTS instrument is currently operational in a solar occultation mode covering a latitudinal range of 85° S to 85° N. It measures temperature, pressure profiles along with concentrations of a number of trace gas species at the upper tropospheric levels to about 150 km. During the retrieval process, the temperature and pressure profiles are retrieved first, which are subsequently used to retrieve the

volume mixing ratios of the atmospheric species. The detailed retrieval algorithm is described in Boone et al., (2005). For this study, we have used the level 2 version 3.5 $CH_4$ data for the year 2010 as a reference for comparison with model $CH_4$ profiles. These data are made available on a 1 km resolution vertical grid ranging from 0.5 km to 149.5 km although the retrieved data are present only at altitudes ranging between 13 and 120 km.

**3. Results**

**3.1 Factors influencing the stratospheric contribution to total column $XCH_4$**

We begin by analysing the spatial distribution of the stratospheric $CH_4$ column abundance and identifying regions where the total column is most sensitive to stratospheric column variability. We compute the pressure-weighted column averaged dry air mole fraction of $CH_4$ using the $CH_4$ fields from the C-IFS model for the year 2010. The profile is then separated into two

parts and the tropospheric and stratospheric partial column averaged mole fractions are computed for which we use the 6-hourly tropopause information from the C-IFS model. Figure 1 shows the column-averaged abundance of $CH_4$ for the stratospheric and tropospheric partial columns as well as the total column for the months June to August, 2010. This figure shows that for the tropical regions, the spatial variability of the total column $XCH_4$ is largely driven by the tropospheric $CH_4$ column abundance, which can be attributed to spatial variability in surface fluxes. In the northern hemisphere, the equator-

to-pole gradient of the stratospheric $CH_4$ column is opposite to that of the tropospheric $CH_4$ column such that the stratosphere acts to smooth the overall tropics-pole gradient in the total column.

Figure 2 shows the variability of the two partial columns and the total column $CH_4$ over the three-month period. We see that the tropospheric column variability is largest around the Tibetan plateau region. The highlands of the Tibetan plateau are

regions of high tropopause variability due to their high elevation (between 3000 and 8848 m above sea level) which cause strong stratosphere-troposphere interaction events like tropopause folds to occur. These events can cause stratospheric air to be transported into the troposphere, which is responsible for the variability of the tropospheric and stratospheric partial column. The tropospheric column variability in this region is as high as 40 ppb while in most other regions of the world the tropospheric $CH_4$ values remain comparatively constant where the variability is less than 15 ppb. The variability in the

tropospheric column is also large for regions that form the $CH_4$ hotspots such as wetlands and rice-growing regions of Bangladesh, India, and China, and anthropogenic emissions, possibly exacerbated by wildfires in 2010, in western Russia.

The stratospheric column variability on the other hand has a zonal distribution. This is because the variability of the stratospheric column is directly linked to the tropopause height (Fig. 3). As expected, the mean tropopause height is higher in the tropics (90-100 hPa) than at extra-tropical and polar latitudes (>150 hPa). In the high- and mid-latitudes, especially in areas at the edge of the southern hemisphere polar vortex, the spatial gradient of the tropopause is at its maximum. The tropopause, therefore, interacts with the jet stream and extratropical weather systems, causing it to move up and down. The vertical movement of the tropopause results in areas of high tropopause height variability during the austral winter months (Fig. 3(b)), which therefore impact the variability in the stratospheric column. During months of boreal winter (not shown), this effect is shifted to the Northern Hemisphere. On the other hand, since the tropical tropopause is rather flat and has a weak spatial gradient, it causes little or no variability in the stratospheric partial column except in the Tibetan highland region (90 ppb).

The impact of the stratosphere on the total column $CH_4$, $XCH_4$, is largely linked to two factors: (i) the mass of $CH_4$ in the stratosphere relative to that in the total column and (ii) its associated variability due to dynamical processes in the atmosphere such as the movement of the tropopause. This means that the contribution of the uncertainties in the stratospheric $CH_4$ to the total column abundance of $CH_4$ is likely to be significant in regions where at least one of the two driving factors is high. For regions where both these factors are low, the $XCH_4$ value is less sensitive to uncertainties in the stratospheric $CH_4$ component. We perform a qualitative analysis of how these two driving factors vary spatially during the different seasons of the year to identify regions where the stratospheric processes directly influence the total column and regions where the impact is not significant.

We define two quantities:

$$CH_4 \text{ mass fraction } (f_{str}) = \frac{\text{mass of CH4 in the stratospheric column (in kg)}}{\text{mass of CH4 in the total column (in kg)}} \tag{1}$$

$$CH_4 \text{ mass fraction variability } (\sigma_{str}) = \text{Standard deviation of } f_{str} \tag{2}$$

In the context of extending the aircraft measured profiles into the stratosphere, it can be said that if an aircraft profile is present in regions having both low $f_{str}$ and low $\sigma_{str}$, the total column is likely to be less sensitive to the choice of data source used as an assumption for the stratosphere. Figure 4 shows the C-IFS stratospheric $CH_4$ mass fraction $f_{str}$ plotted against its variability $\sigma_{str}$ for five different latitude bands during the different seasons. It can be seen that overall, the tropics are regions with both low $f_{str}$ and low $\sigma_{str}$ throughout the year, while the extra-tropical and high-latitude regions have high values for either one or both of these factors, making the computed value of the total column in these regions more sensitive to the $CH_4$ variability in the stratosphere. During the austral winter months, the Southern Hemisphere shows particularly high variability in the stratospheric $CH_4$, which is likely to be due to the impact of the polar vortex dynamics.

The latitudinal distribution of airports visited by the MOZAIC fleet during one year (2004), reflecting the typical yearly MOZAIC flight statistics shows that while almost all the profiles are measured in the Northern Hemisphere, they are mostly concentrated in the mid-latitude region (between 40° N and 55° N). This is because of the large air traffic between Europe and North America by the airlines participating in MOZAIC. Of all the MOZAIC profiles measured in one year, only a small fraction falls within the tropical region (about 17 %). It is thus reasonable to infer that for the passenger aircraft profiles with sampling comparable to MOZAIC, the stratospheric variability is critical to determining the total column $CH_4$ abundance and needs to be accounted for using an appropriate method of profile extension into the stratosphere.

In the following sections, we compute and compare the uncertainty introduced in the total column at the MOZAIC airport locations using the model output, climatology data and a-priori profile as stratospheric extensions.

## 3.2 C-IFS and TOMCAT models

We compare the model profiles from C-IFS and TOMCAT models to coincident satellite observations from MIPAS. These
measurements are independent since these are not assimilated into the models. The 6-hourly model profiles are interpolated to the time and location of the satellite observed soundings - linear in time and closest neighbour in space. The MIPAS profiles are then interpolated onto the coarser model vertical grids. We do not apply averaging kernel information to the coincident model profiles since the impact is not expected to be significant (Laeng et al., 2015; Ridolfi et al., 2011). In order to make a true comparison between the stratospheric levels of the profile simulated by the two models we use the C-IFS
tropopause height for identifying and analysing the stratospheric levels for the TOMCAT model. Because the TOMCAT model is driven by winds from ERA-Interim, this definition of the tropopause height should be consistent with the transport of TOMCAT.

Comparison of zonal mean model profiles and coincident satellite observations for the months September to November is shown in Fig. 5 and 6. We see that the C-IFS is biased high compared to the observed value from MIPAS in the lower
stratosphere just above the tropopause (at around 100 hPa) by about 80-100 ppb during the months of September to November (Fig. 5(d)). This bias reverses in sign and increases to about 200 to 300 ppb in the middle stratosphere (10 hPa pressure level). In the tropical latitudes this bias shifts to the upper layers of the stratosphere (around 1 hPa). Furthermore, a comparison between Fig. 5(a) and 5(b) shows that the C-IFS model simulates a steeper vertical gradient in the $CH_4$ concentration in the stratosphere as compared to that observed by MIPAS.

The comparison between TOMCAT and MIPAS for the same period shows that TOMCAT is biased high by about 100 ppb compared to the MIPAS soundings in the lower stratosphere (100 hPa). In the middle stratosphere (10 hPa) the bias reverses in sign (-100 to -200 ppb in the Southern Hemisphere and around -50 ppb in the Northern Hemisphere mid-latitudes) and again becomes positive (~100 ppb) in the upper stratospheric layers. Thus, the positive and negative bias patterns in the stratospheric levels occur alternately. Also the gradient in the $CH_4$ concentration in the stratospheric levels as simulated by
TOMCAT is more comparable to the observations and is not as steep as that modelled by C-IFS.

In order to further investigate the spatial patterns of the stratospheric bias, we evaluate the satellite observed $CH_4$ concentrations and the models sampled at the locations of the satellite measurements at a given pressure level. We chose 10 hPa, since the observed biases are highest around this pressure. From Fig. 7(c), 7(d) and 8(c), 8(d), we find that for both instruments, the bias in the C-IFS model forms zonal bands with little variability. Since the data density from MIPAS is
much higher, these patterns are more clearly seen in Fig. 7. From Fig. 7(e) and 8(e) we see that the TOMCAT model bias in the middle stratosphere with reference to the two satellite instruments compare well with each other, with the highest bias during Sep-Nov 2010 being around the North Pole (~400 ppb). The spatial distribution of the bias is not quite as zonal as is seen in the C-IFS and is more irregular in structure. This difference in the bias pattern between the two models can be attributed to the fact that the TOMCAT simulation used here fails to capture the observed zonal structure of the $CH_4$
distribution (Fig. 7(c)) while the C-IFS does a much better job at simulating the longitudinal patterns (Fig. 7(b)) in the satellite data from MIPAS or ACE-FTS measurements.

A similar comparison was made for the two models for the other seasons of the year (not shown) and it was seen that these

biases are a constant feature throughout the year with the magnitude and distribution being almost the same for all seasons. We also compared the $CH_4$ profiles from the ACE-FTS instrument and the C-IFS fields to investigate if the biases obtained by comparison with MIPAS are in agreement (Fig. 6). Although MIPAS has much better data coverage than ACE-FTS, with measurements made at all latitudes and the number of MIPAS profiles measured per day being significantly larger than those measured by ACE-FTS, we find that the model bias as observed by ACE-FTS is similar in magnitude and distribution to that observed by MIPAS and the two comparisons are in good agreement with one another. The $CH_4$ gradient in the vertical as observed by ACE-FTS is also much shallower than that simulated by C-IFS, a feature consistent with that seen by MIPAS.

We further compute the column-averaged bias for the stratospheric levels in the C-IFS and TOMCAT models. Comparing the bias allows us to evaluate the sources of model error in the stratospheric extension of aircraft profiles. Here, we make an implicit assumption that the aircraft profiles reach the altitude of the tropopause and that the entire column above the tropopause height is unmeasured and has to be extended artificially using the model data. Since the MIPAS instrument offers the advantage of more complete global coverage over ACE-FTS, we use it as our reference for the subsequent analysis of stratospheric column bias. We compute the overall bias in the stratospheric column by carrying out a mass-weighted integration of the bias in each model with respect to the satellite soundings from MIPAS for each pressure level above the tropopause. We restrict our analysis to only those latitudes where the aircraft profiles are likely to be measured, i.e. we do not consider the latitudes poleward of 60° S and 80° N. Thus, we exclude the polar regions, over which no commercial aircraft are likely to fly and it is reasonable to exclude those latitudes from the analysis for the purposes of this study.

Figure 9 shows the zonally averaged stratospheric column bias relative to MIPAS for C-IFS and TOMCAT. We see that the overall absolute magnitude of the bias in the stratospheric column of the C-IFS is less than 15 ppb. This bias translates to less than 1 % of the total column $CH_4$ abundance. The bias magnitude changes with season and latitude. Overall, in the Northern Hemisphere the bias is lowest during the autumn months (SON) and highest in spring (MAM). The opposite is observed in the Southern Hemisphere. The errors in the Southern Hemisphere could be partly due to the inability of the model to capture the dynamics of the polar vortex and the extra-tropical storm track that develops in the Southern Ocean during autumn-winter months. These are associated with tropopause folds in the development of synoptic weather systems which are generally not as well captured as those in the northern hemisphere due to a sparser observing system (Bauer et al. 2015; Haiden et al. 2015). The summer and winter bias values lie intermediate to the spring and autumn bias globally. The zonal mean bias in TOMCAT has a similar seasonally- and latitudinally-varying nature as C-IFS albeit with a smaller magnitude. The bias throughout lies between ±5 ppb, which translates to 0.2 % of the total column value, which is much smaller than the C-IFS model bias. This is likely to be due to the fact that these values are averages over all longitudes and, therefore, any variation in the bias along the longitude will be smoothed out.

We further analyse the stratospheric column bias at actual aircraft profile locations to get a realistic estimate of the bias from both models. For comparison, we use MIPAS profiles measured on the same day as the aircraft profiles and within ±2° longitude and latitude in space. We find that for real aircraft profile locations, both models have the same mean bias (about -9 ppb) in the stratosphere (Fig. 10, Table 1). The C-IFS bias however has a higher precision (standard deviation of 6.5 ppb) compared to TOMCAT (standard deviation of 12.8 ppb). As per the random error (precision) and systematic error (accuracy) requirements specified in Buchwitz et al., 2011, the errors from both models are lower than the minimum ('threshold') accuracy and precision requirements for XCH4. In addition, the C-IFS model random error also meets the targeted precision ('goal') requirement (9 ppb).

### 3.3 Climatology-based approaches

We now explore the potential of climatology-based approaches as stratospheric extensions for the aircraft profiles that, for instance, could be based on balloon-based measurements, satellite limb soundings or those from AirCore. Climatology based measurements are typically long term averages having a much sparser global coverage compared to global model output. For this part of the study, no real observations are used and we only evaluate the contribution of sparse data coverage and temporal averaging to the stratospheric column uncertainty. In order to do this, we analyse two main cases:

1. mmC-IFS: In this case, we use monthly mean C-IFS fields for our stratospheric assumption instead of full C-IFS fields with 6-hourly output (the FULL C-IFS case). This means that we do not account for the synoptic scale variability in the $CH_4$ vertical distribution. This helps us examine the impact of temporal variability of the data source on the stratospheric column bias.

2. In addition to the temporal variability, we test the impact of reduced spatial coverage of the data source for the stratosphere. We use the C-IFS $CH_4$ fields sampled at measurement locations from two satellite instruments:
(a) mmC-IFS@ACE-FTS: Full C-IFS $CH_4$ fields are sampled at the ACE-FTS measurement locations, after which monthly means are obtained and interpolated to obtain global fields at C-IFS resolution.
(b) mmC-IFS@MIPAS: Similar to 2(a), using sampling locations and time from the MIPAS instrument.

Comparison of the above three scenarios with the FULL C-IFS case helps draw conclusions about how well the stratospheric column can be captured with limited temporal and/or spatial coverage of the data. Since the MIPAS instrument has much better coverage than ACE-FTS, we expect the fields obtained from mmC-IFS@MIPAS to be closer to the truth (in this case FULL C-IFS) than mmC-IFS@ACE-FTS. The idea here is to not compare the two instruments but evaluate the impact of high/low data coverage in addition to reduced temporal variability. We analyse the histograms (Fig. 11) showing the stratospheric column bias and its variability for each of the above cases with respect to FULL C-IFS and subsequently convert these to values with MIPAS as a reference (Table 1). This is done by adding the bias in the FULL C-IFS with respect to MIPAS to the bias values computed for each of the scenarios. The random error or standard deviation is converted by computing the square root of the sum of the variance in the FULL C-IFS and that from each case.

We find that the mean bias increases slightly to -14 ppb in the case where only monthly mean fields from C-IFS (mmC-IFS) are used, and increases to -32 ppb in mmC-IFS@ACE-FTS. The variability increases strongly to 49 ppb and 200 ppb for the two cases. In mmC-IFS@MIPAS, the mean reduces to 3 ppb which is better than the mean bias in the FULL C-IFS (-9 ppb). However, since the variability in the stratospheric column error is still about 10 times larger than that of the FULL C-IFS (around 57 ppb), it cannot be deemed fit for estimating the stratosphere well. As expected mmC-IFS@ACE-FTS performs poorly as compared to mmC-IFS@MIPAS both in terms of the bias and variability, owing to the fact that the monthly sampling from ACE-FTS is much sparser than that of MIPAS.

### 3.4 Satellite a-priori profile

Finally, we evaluate the possibility of using a-priori profiles used in satellite data retrievals to extend aircraft profiles into the

stratosphere. For this, we use the University of Leicester GOSAT Proxy XCH$_4$ retrieval (Parker et al., 2011). The a-priori profile used in this retrieval is based on a CH$_4$ simulation using the TM3 transport model. Figure 12 shows the distribution of the stratospheric column bias at the MOZAIC airport locations, with respect to collocated MIPAS CH$_4$ profiles. We see that the mean error in the stratospheric column is about -14.7 ppb while the random error amounts to 53 ppb (Table 1). These

values are comparable to those obtained from the mmC-IFS case in Sect. 3.4 but are still much higher than the bias and random error obtained from the C-IFS and TOMCAT models.

## 4 Discussion and conclusions

The suitability of airborne measurements as reference data for the validation of satellite based total column measurements is
well documented. Previous studies have shown that the unsampled part of the atmosphere above the aircraft ceiling contributes the largest uncertainty in the total column computed from aircraft profiles (Wunch et al., 2010). In this study, we analyse three different stratospheric CH$_4$ data sources that can be used for the purpose of aircraft profile extension by comparing the bias each data source introduces in the total column. For realistic bias estimation, the value of the bias is computed at the location of the MOZAIC airports.

Our results show that the C-IFS and TOMCAT models show smaller biases and standard deviation values of the stratospheric column error at the airport locations than those computed using scenarios that simulate the use of climatology datasets and the satellite a-priori profile. While the bias from both the models in the stratosphere is about -9 ppb, the random error in the C-IFS is smaller in magnitude (6.5 ppb) than that from the TOMCAT model (12.8 ppb). These values are within the minimum requirements for total column CH$_4$ retrievals from satellites as specified in Buchwitz et al., (2011). The error
from the C-IFS model, additionally meets the 'goal' or targeted requirement. Application of latitudinal and seasonal bias correction to the model fields is likely to produce even better results. We need to keep in mind that while both models seem to be performing equally well in the stratosphere there are significant differences in the datasets from the two models in terms of how they are generated. The C-IFS is a data assimilation model that simulates tropospheric CH$_4$ in detail. However, since the model initial conditions are constrained by the assimilated observations for NWP, its use could be circular. In
addition, the stratospheric chemistry used in the model is parameterized. On the other hand, TOMCAT is a chemical transport model that is driven by the ERA-interim meteorology. The treatment of tropospheric CH$_4$, however, is simplified in the model. The TOMCAT model improves over the C-IFS model due to the realistic treatment of stratospheric sinks, which is reflected in the lower mid-stratospheric bias (-100 to -200 ppb) in comparison to the C-IFS analysis (200 to 300 ppb). In other words the TOMCAT results show that ongoing developments to include a more realistic implementation of
stratospheric chemistry in C-IFS should improve the bias relative to the satellite observations. In addition, the C-IFS model output used here is at a higher horizontal resolution than TOMCAT (approximately 0.8° compared to 2.8°), which is also likely to impact the bias. This can be improved by running the TOMCAT model in a different configuration. It is worth mentioning that since the C-IFS is optimised in the troposphere, unlike the TOMCAT simulation used here, it can also be used as reliable extension for any tropospheric levels that are not measured by the aircraft.

We further investigate the impact of reduced synoptic scale variability and spatial coverage of the data source used for stratospheric extension, such as when using a climatology. We find that the spatial coverage of the data source impacts the bias greatly, as is clear in the case of mmC-IFS@ACE-FTS (-32 ppb bias, 200 ppb standard deviation) as compared to mmC-IFS@MIPAS (3 ppb bias and 56.7 ppb standard deviation) since the ACE-FTS instrument has poorer spatial coverage

compared to the nearly global coverage by MIPAS. It should be noted that the evaluation of the MIPAS and ACE-FTS instruments in this section is only a theoretical exercise to evaluate the influence of spatial sampling and coverage in estimating the bias. In any case, the MIPAS instrument is no longer operational and cannot be used as a potential stratospheric extension data source while ACE-FTS, though currently operational, might not work for much longer (SCISAT-1 has long surpassed its expected lifetime of two years). Hence, other limb sounding instruments measuring trace gas profiles in the UTLS region are needed in the coming years. This analysis also highlights the shortfalls of any climatology based on sparse profile measurements such as those from balloons or AirCore. Lastly, on using the GOSAT a-priori profile for profile extension, we find that the resulting stratospheric uncertainty is comparable to the case where monthly mean C-IFS fields are used. However, the random error in this case is much higher than the case where full fields from the C-IFS model are used making the a-priori profile a less favourable option among other data sources considered in this work.

In summary, our work offers insights into the different data sources that can be used for the purpose of completing the "missing" part of the $CH_4$ profile from aircraft when using these profiles for satellite validation. We demonstrate that using bias-corrected model fields are likely to produce the best results in the stratosphere for $CH_4$. In-situ profiles from balloon borne AirCore measurements can prove to be very useful in this regard. These profiles extend up to an altitude of about 30 km and can be good sources of reference data for model validation and bias correction in the UTLS regions. In the coming years, an increased number of aircraft profiles of greenhouse gases, for instance, those from the IAGOS project, are expected to be available. Besides having great potential for providing robust validation methodologies of remote sensing observations and atmospheric models, these measurements have applications in NWP (e.g. in bias correction schemes or for data assimilation) as explored by the CAMS system. This can go a long way in contributing to an integrated global observing system and providing deeper insights into the chemical and physical processes in the atmosphere.

**Acknowledgements**

We thank Wuhu Feng (Leeds) for help with the TOMCAT model, which was supported by NCAS.

We thank the KIT-IMK team for making the MIPAS methane data available to us.

The Atmospheric Chemistry Experiment (ACE), also known as SCISAT, is a Canadian-led mission mainly supported by the

Canadian Space Agency and the Natural Sciences and Engineering Research Council of Canada. We thank the ACE-FTS science team for providing methane data for this study.

The research leading to these results has received funding from the European Community's Seventh Framework Programme ([FP7/2007-2013]) under grant agreement n° 312311 for the IGAS project (IAGOS for the GMES Atmospheric Service).

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

|  | MEAN BIAS (ppb) | VARIABILITY (ppb) |
|---|---|---|
| *Model output* |  |  |
| C-IFS | -9.0 | 6.5 |
| TOMCAT | -9.1 | 12.8 |
| *Climatology-based approaches* |  |  |
| mmC-IFS | -14.2 | 49.0 |
| mmC-IFS @ MIPAS | 3.0 | 56.7 |
| mmC-IFS @ ACE-FTS | -32.0 | 200.0 |
| *GOSAT a-priori profile* | -14.7 | 53.0 |

**Table 1: Mean value and variability of the stratospheric column bias due to the different stratospheric extensions at the locations of MOZAIC airports. MIPAS is taken to be the reference truth. The documented 'threshold' requirements of bias/systematic error (as a measure of accuracy) and random error (as a measure of precision) for satellite based XCH$_4$ to be usable for CH$_4$ source/sink estimation are 10 ppb and 34 ppb respectively (Buchwitz et al. 2011)**

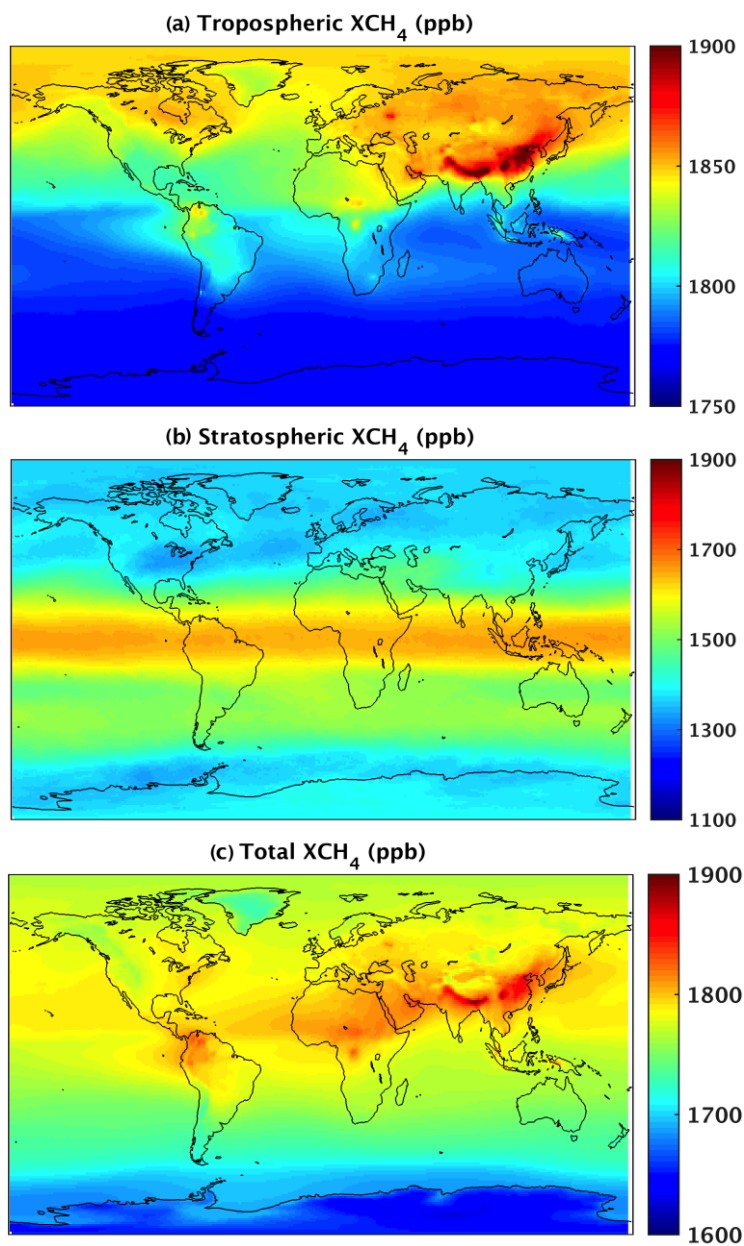

**Figure 1: Mean column abundance of CH₄ (in ppb) during June-August 2010 obtained from the C-IFS fields for (a) tropospheric partial column, (b) stratospheric partial column and (c) total column. Note the different colour scales in the three panels.**

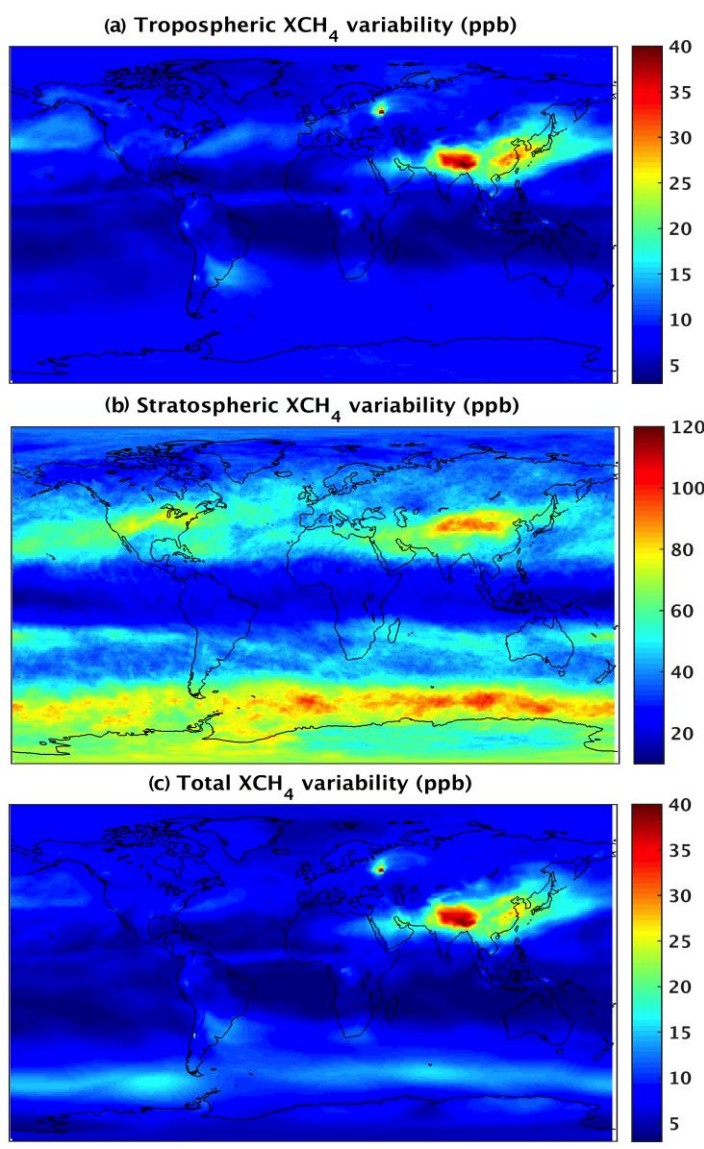

**Figure 2: Variability (standard deviation) in the column abundance of CH₄ (in ppb) during June-August 2010 obtained from the C-IFS model fields for (a) tropospheric partial column, (b) stratospheric partial column and (c) total column. Note the different colour scales in the three panels.**

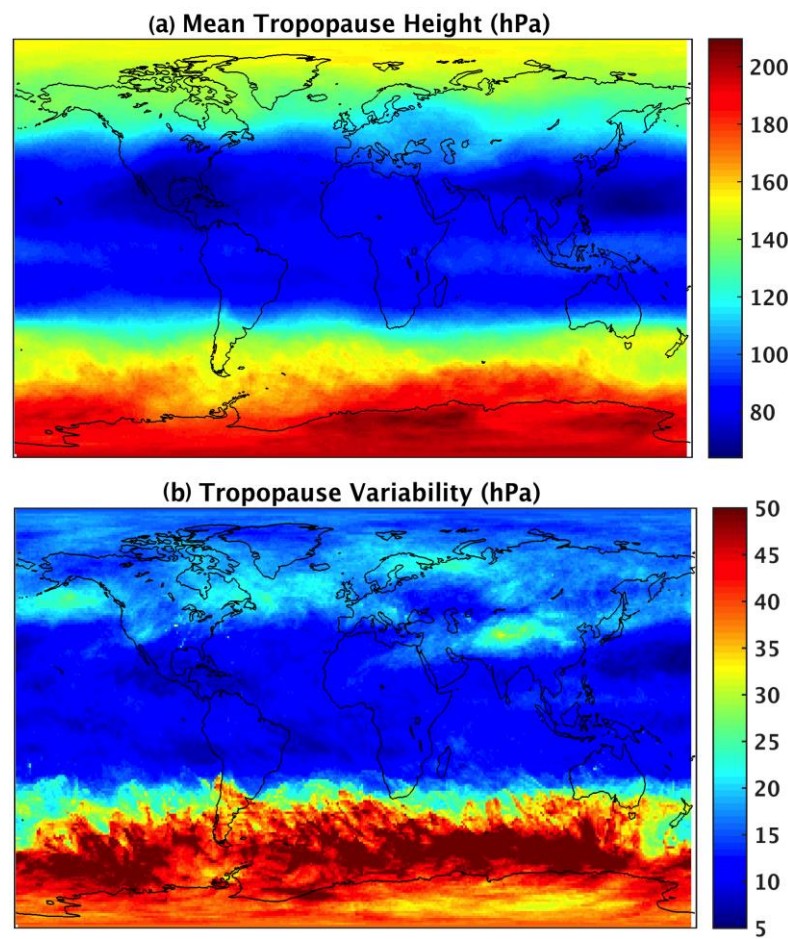

**Figure 3: (a) Mean tropopause height (in hPa) and (b) variability (standard deviation) of tropopause height (in hPa) from the C-IFS model fields for June - August 2010.**

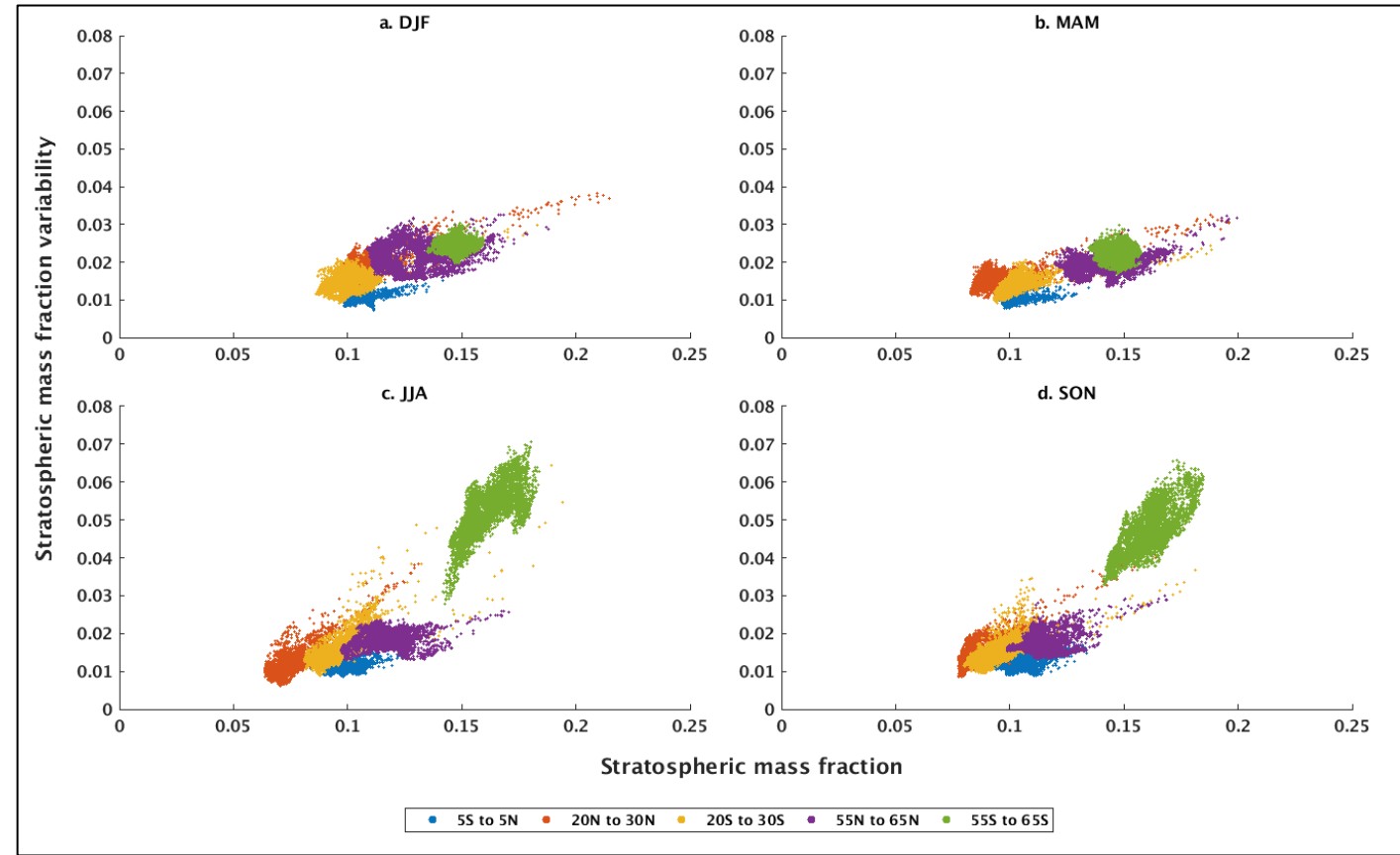

**Figure 4: Scatterplots showing the CH₄ stratospheric column mass fraction ($f_{str}$) against CH₄ stratospheric column mass fraction variability ($\sigma_{str}$) for (a) December-February, (b) March-May, (c) June-August and (d) September-November months of 2010. The colour shading indicates different latitude bands.**

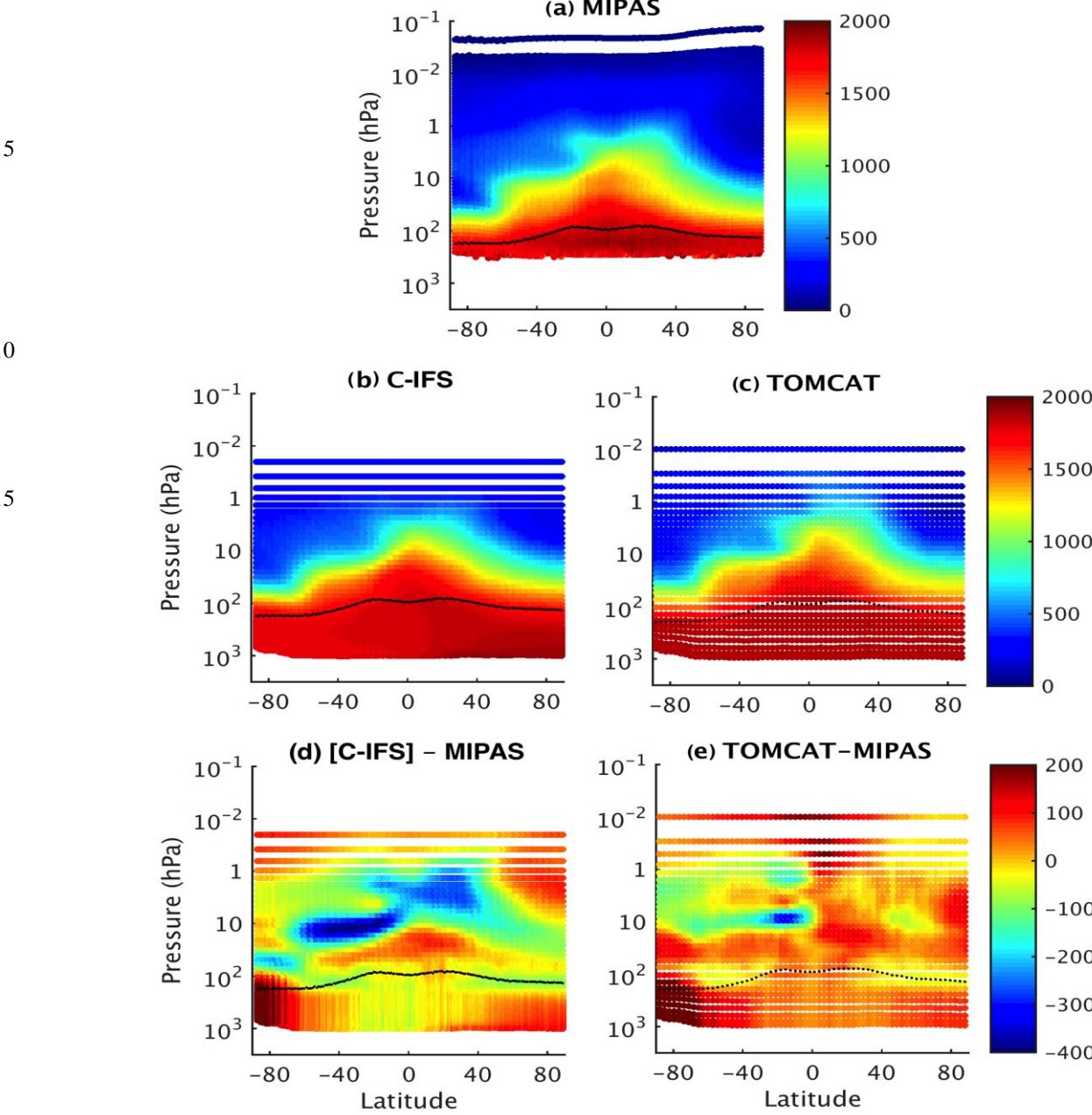

**Figure 5: Zonal mean latitude-pressure plots of CH$_4$ (in ppb) for the months September to November 2010. Panel (a) shows the profiles from the MIPAS satellite soundings. Panels (b) and (c) show the profiles from the C-IFS and TOMCAT models, respectively, sampled at the location and time of the MIPAS measurement. Panels (d) and (e) show the bias between the models and MIPAS measurements. The tropopause location is shown as black dots.**

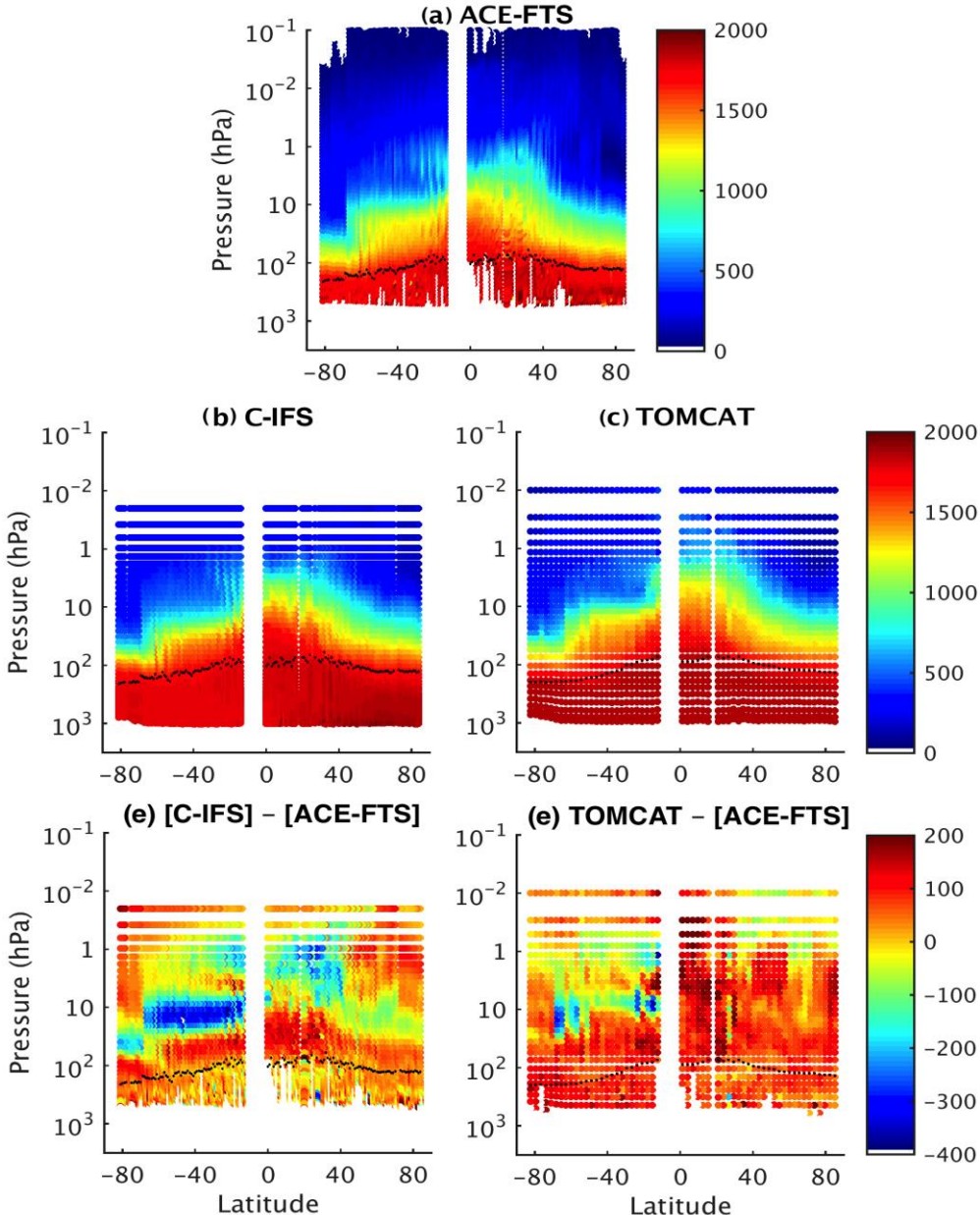

**Figure 6: Zonal mean latitude-pressure CH₄ profiles (in ppb) for the months September to November 2010 plotted against latitude. Panel (a) shows the profiles from the ACE satellite soundings. Panels (b) and (c) show the profiles from the C-IFS and TOMCAT models, respectively, sampled at the location and time of the ACE measurement. Panels (d) and (e) show the bias between the models and ACE measurement. The tropopause location is shown as black dots.**

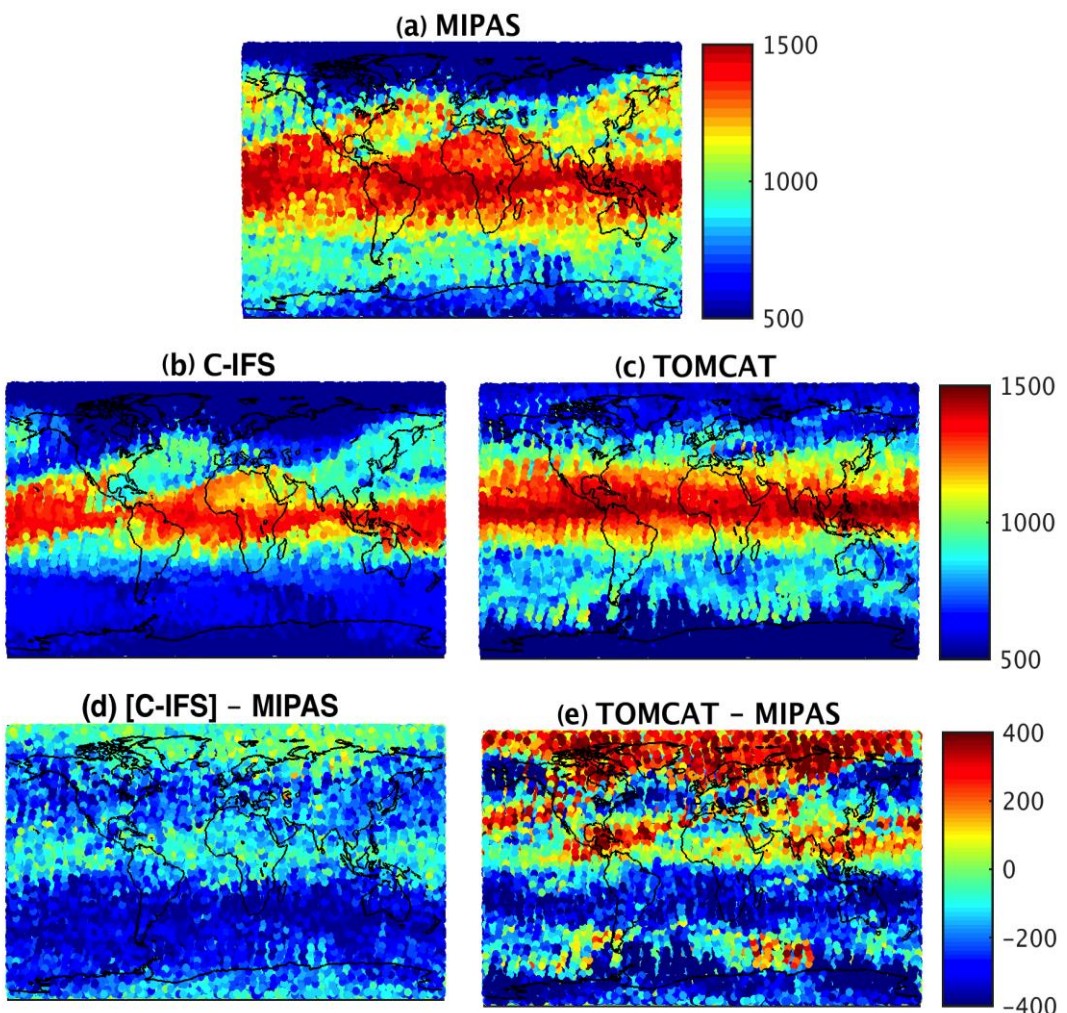

**Figure 7: Maps showing the CH₄ concentration (in ppb) at the 10 hPa pressure level for the months September to November 2010. Panel (a) shows the CH₄ concentration as measured by MIPAS. Panels (b) and (c) show the CH₄ concentrations modelled by C-IFS and TOMCAT, respectively, sampled at the location and times of the MIPAS measurements. Panels (c) and (d) show the bias between the models and the MIPAS measurements.**

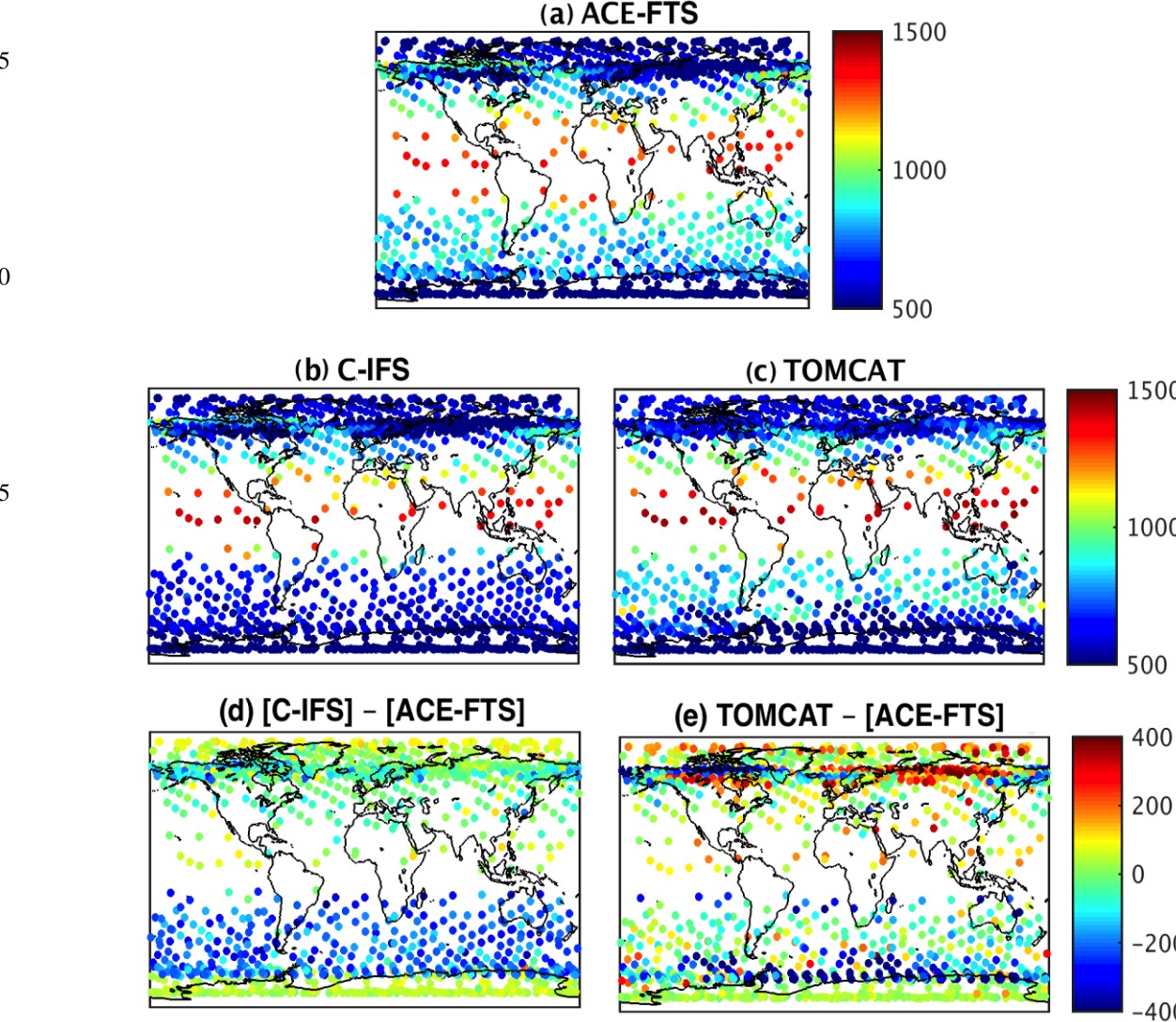

**Figure 8: Maps showing the CH₄ concentration (in ppb) at the 10 hPa pressure level for the months September to November 2010. Panel (a) shows the CH₄ concentration as measured by ACE. Panels (b) and (c) show the CH₄ concentration modelled by the C-IFS and TOMCAT, respectively, sampled at the location and times of the MIPAS measurements. Panels (c) and (d) show the bias between the models and the ACE measurements.**

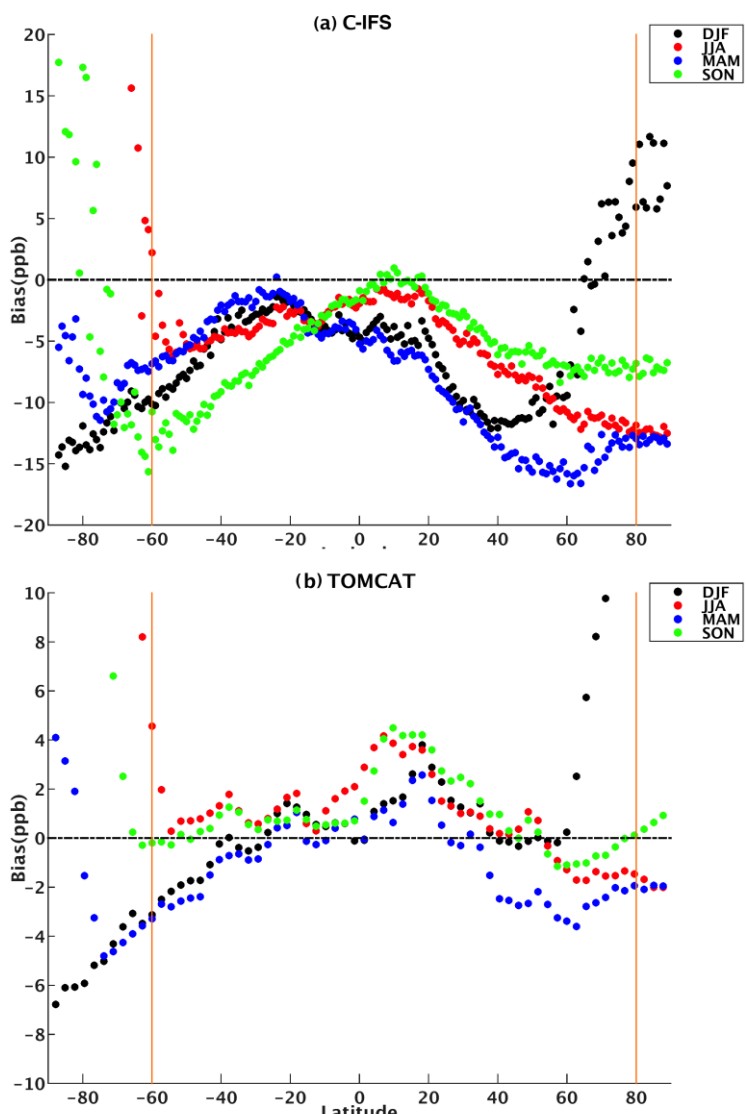

**Figure 9: Zonal mean CH₄ stratospheric column bias for different seasons of the year 2010 plotted against latitude for the models (a) C-IFS and (b) TOMCAT. MIPAS data are used as reference truth. Note the difference in the scaling of the y-axis.**

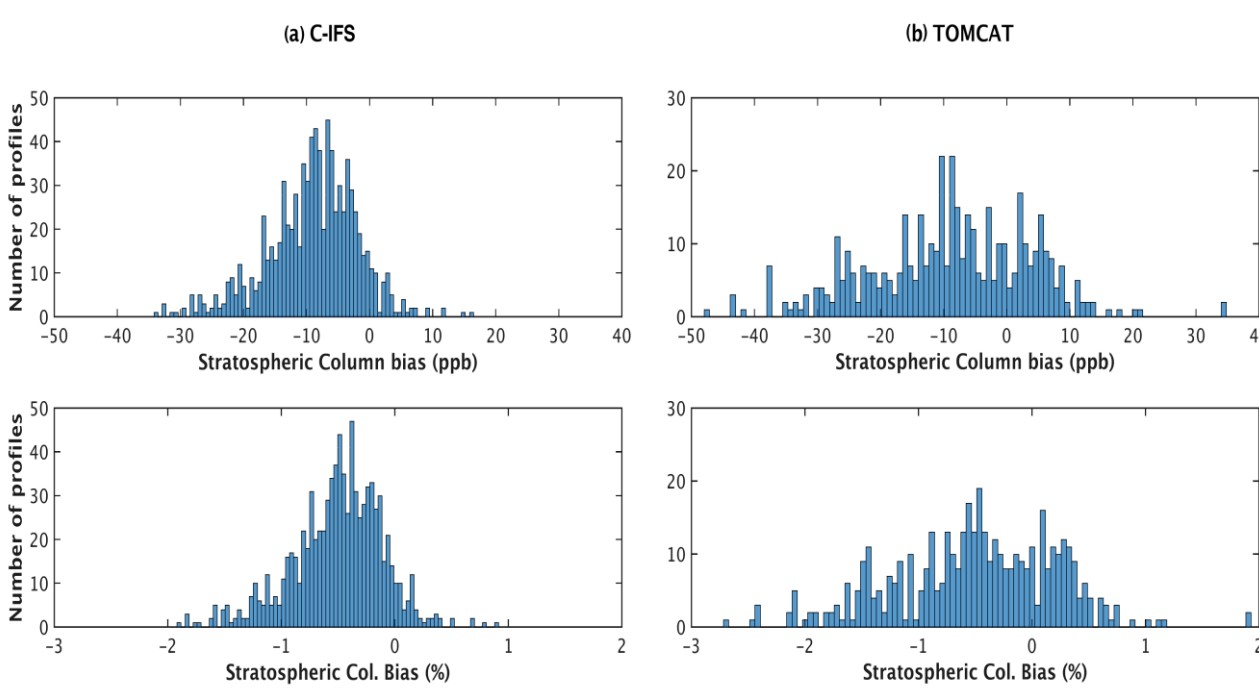

**Figure 10: Histograms showing the distribution of the stratospheric column bias with respect to MIPAS at the MOZAIC airport locations for the year 2010 for (a) C-IFS model and (b) TOMCAT model.**

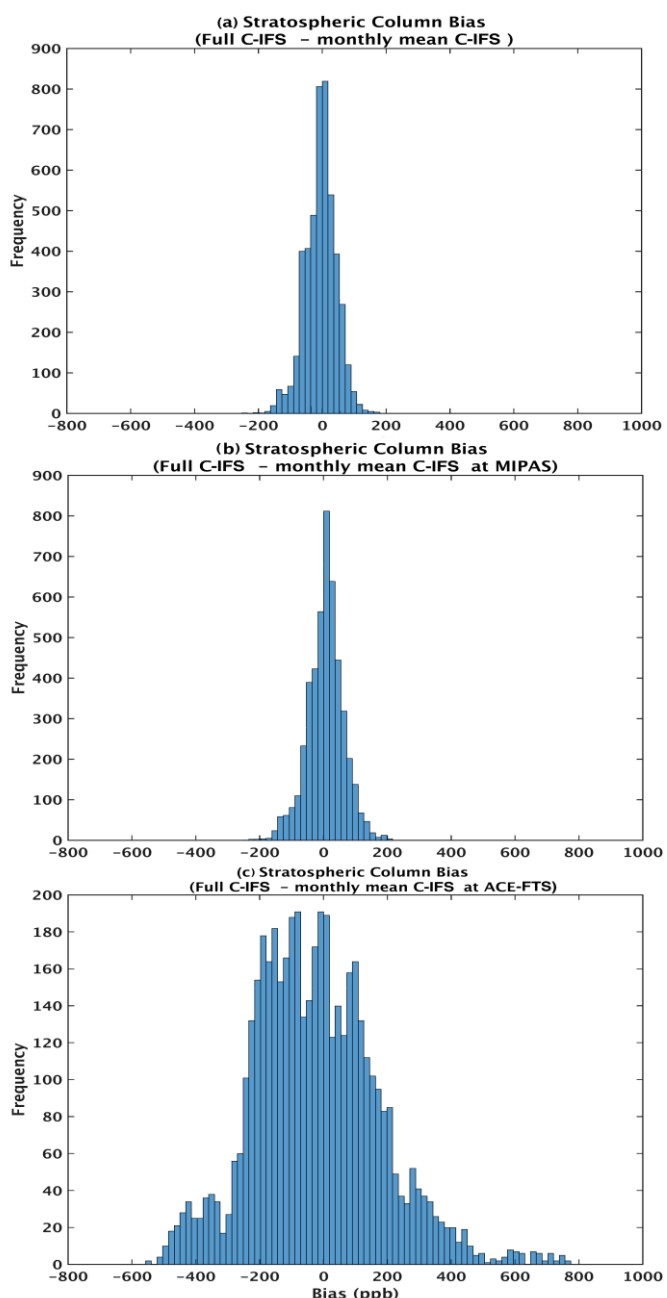

**Figure 11: Distribution of the stratospheric column bias estimated at the location of the MOZAIC airports and using FULL C-IFS as the reference truth. Panel (a) shows the bias when monthly mean fields from the C-IFS model are used for profile extension. Panels (b) and (c) depict the bias when monthly mean fields from the C-IFS model obtained using the sampling from the MIPAS and ACE instruments are used for profile extension, respectively.**

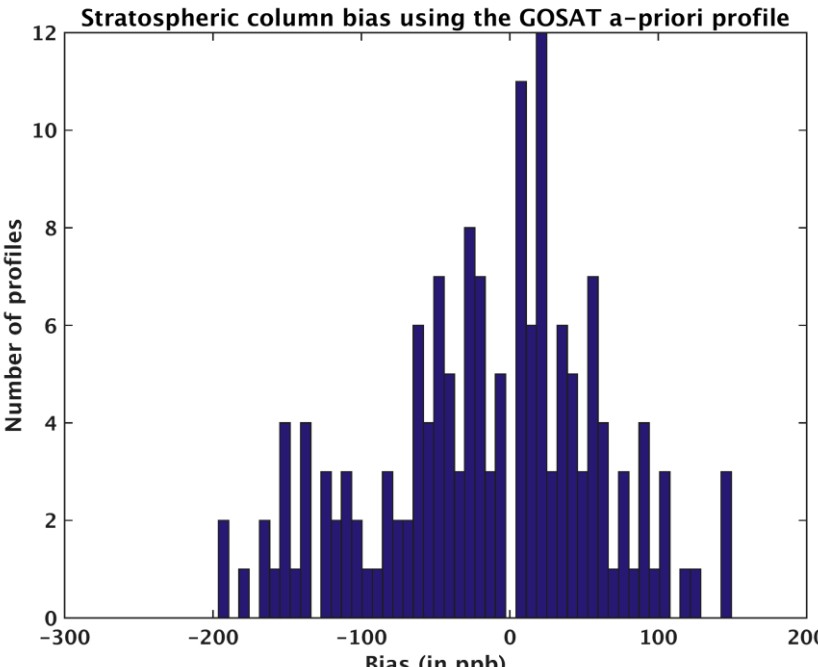

**Figure 12: Stratospheric column error estimated at the MOZAIC airport locations when the GOSAT CH$_4$ a-priori profile is used for aircraft profile extension. MIPAS data are taken as reference truth.**