# Peer review of "Extending methane profiles from aircraft into the stratosphere for satellite total column validation using the ECMWF C-IFS and TOMCAT/SLIMCAT 3D model"

_Atmospheric Chemistry and Physics, 2016_

## Referee Comment (RC1) · Anonymous Referee #1 · 14 Nov 2016

The study is well written and addresses an important topic that falls under the scope of ACP. I advice to publish this paper after major revisions.

The study uses the ECMWF C-IFS and TOMCAT/SLIMCAT 3D models for the extension of MOZAIC/IAGOS airborne CH4 profiles into the stratosphere. This is required to calculate more accurate vertical column densities that fulfill the precision and accuracy demands of modern satellite missions. An exemplary comparison with MOPITT and ACE-FTS CH4 vertical columns is performed.

The title of the paper states that the study performs a comparative analysis of different

data sources, however this is missing in the manuscript. More precisely, section 3.3 about climatology based approaches is again only an application of the ECMWF C-IFS and TOMCAT/SLIMCAT model for different spatial and temporal samplings and section 3.4 about satellite a priori profiles is much too short with 7 lines to justify this title.

Major changes:

- I suggest to change the title of the work to "Extending methane profiles from aircraft into the stratosphere for satellite total column validation using the ECMWF C-IFS and TOMCAT/SLIMCAT 3D model.".

- p3,l20 - p4,l1-12: This text fragment is strongly overlapping with the following section 2. Please shorten this part and remove the information repeated in section 2.

- p9,l36 -p10,l3: It is not clear how this bias correction is actually done. This text fragment must be improved or removed together with figure 12.

- p8,l35-l36: It is crucial to show that this bias is constant for other seasons of the year. Hence, this must be discussed in more detail (plots, statistics)

- p18, table 1: I don't get how the GOSAT a-priori profile can outperform the mmc-IFS@MIPAS for the variability and mmc-IFS@ACE-FTS for mean bias and variability and even reaches the same performance as the mmC-IFS case. What does that mean? Can you comment on that?

- figure 8: I think figure 8 and the related discussion in the text would be more meaningful when considering the stratospheric partial column instead of only one level at 10 hPa.

Minor changes:

- p1, l34-l38 please name here further satellite missions that are aiming on CH4 and CO2 vertical column measurements and not only GOSAT.

- The panels (a), (b) from figure 6,7,8,9 can be removed the difference to the reference

is shown so absolute values are redundant information.

- figure 5 can be removed, the information content is to low and is already described in the text.

- I would like to see figure 1,2 in the unit molec/cm2 this would allow to relate the partial columns to the total column.

- figure 4 must be improved. The most interesting latitude range 30 - 55 degree is covered by the other data and cannot be seen in this representation.

[Figure]

---

## Referee Comment (RC2) · Anonymous Referee #2 · 25 Jan 2017

The paper provides a valuable study on extending aircraft measurements to stratosphere for validating satellite total column retrievals. Overall it is well-written, and the results are useful. It should be accepted for publication after minor revisions.

General comments:

1. As the MIPAS data are used as 'the reference' to quantify bias and random errors of the (extended) stratospheric CH4 data, a more detailed about the quality of MIPAS data is helpful. Also, the errors of the MIPAS data should be taken into account when the authors discuss whether the precision and accuracy of the resulting stratospheric CH4 columns meet the requirement for satellite XCH4 validations etc.

2. I'd like some discussions on using the extended aircraft data for validating CH4 retrievals from other instruments such as IASI, particularly over extra-tropical regions
Minor comments:

1. Line 20, Page 5: '..were specified from monthly global mean observations.. Please specify which CH4 observations were used to construct the global monthly mean'

2. Line 25, Page 5:'..which has no additional constraint...' To be more accurate, please add 'in stratosphere'

3. Line 27, Page 7:'....stratospheric CH4 that is likely to be due to the impact of the polar vortex dynamics',

Change to '...stratospheric CH4, which is likely to be due to the impact of the polar vortex dynamics'

4. Line 27, Page 9: '...therefore, any variation in the bias along the latitude will be smoothed out', Why averaging along longitude will 'smoothing out any variations along the latitude' ?

5. Line 37, Page 9:'...We test this by applying a bias-correction to the ..', More details about bias correction is helpful.

---

## Author Comment (AC1) · 31 Mar 2017

The comment was uploaded in the form of a supplement:
http://www.atmos-chem-phys-discuss.net/acp-2016-704/acp-2016-704-AC1-
supplement.pdf

---

## Author Response (AR1)

**Authors' responses to reviewers' comments:**

We would like to thank both the referees for their careful reviewing and constructive comments and suggestions for this manuscript. Our responses to the comments are as follows:

**[RC]:** Reviewer's comment **[AR]:** Authors' response **[ME]:** Manuscript edits & modification

5 **Reviewer # 1**

[RC] - I suggest to change the title of the work to "Extending methane profiles from aircraft into the stratosphere for satellite total column validation using the ECMWF C-IFS and TOMCAT/SLIMCAT 3D model.".
[AR]: Done

10 [RC] P3L20 to P4L1-12: This text fragment is strongly overlapping with the following section 2. Please shorten this part and remove the information repeated in section 2.
[AR]: Done
[ME]: "The model output analysed in this study is obtained from two models:
1. The Integrated Forecasting system for Composition (C-IFS) (Flemming et al., 2015; Massart et al. 2014), a
15 comprehensive, state of the art numerical weather prediction (NWP) and Earth-system model developed at the European Centre for Medium - Range Weather Forecasts (ECMWF). 2. The TOMCAT/SLIMCAT model (Chipperfield, 1999; 2006), a 3-D offline chemistry transport model that simulates the temporal and spatial distribution of chemical tracers in the troposphere and stratosphere

As a sanity check, we also compare the model bias to that obtained using $CH_4$ profiles from the ACE-FTS
20 instrument (Bernath et al., 2005) on the Canadian satellite SCISAT-1.

Since climatology-based data are long-term averages, generally with sparse spatial coverage, we further investigate the impact of using these data for the stratosphere by simulating the effect of temporal averaging and reduced spatial coverage on the stratospheric column error. For this, we analyse the error introduced by the following: 1) Monthly mean $CH_4$ fields from the C-IFS model. 2) Monthly mean C-IFS fields based on sampling
25 as that of the (a) ACE-FTS and (b) MIPAS instruments for the stratosphere. This helps to quantify how much uncertainty is introduced if there is a poorer representation of the $CH_4$ variability in the data and if the spatial coverage of the data is low. Further, it allows us to determine if it is better to use the full variability of $CH_4$ from a (potentially biased) model rather than the lower-bias monthly means lacking temporal variability from mean satellite fields. It is noteworthy that the idea behind option 2) is to not compare the impact of using the profiles
30 from the two instruments per se, since MIPAS is no longer flying and hence cannot be used for profile extension in the future, but to evaluate the effect of the different type of sampling from the two instruments i.e. ACE-FTS-like (sparse) and MIPAS-like (dense). Since there is no realistic "truth" of MIPAS or ACE measurements at all times and all places throughout the month, here the full C-IFS fields are treated as the truth and compared to monthly mean fields derived from the C-IFS sampled at the MIPAS and ACE-FTS locations and times. Thus, for

this part of the study, no actual climatology data are used and only the uncertainty introduced by the sampling and averaging is assessed. The computed error in the two cases is then re-calculated with respect to MIPAS using the bias in the full C-IFS fields obtained from comparison with MIPAS.

Lastly, the stratospheric column uncertainty from using the a-priori profile of the satellite retrieval for profile extension is estimated. This is achieved using the University of Leicester GOSAT Proxy $XCH_4$ retrieval (Parker et al., 2011). "

[RC] P9L36 -P10L3: It is not clear how this bias correction is actually done. This text fragment must be improved or removed together with figure 12.
[AR]: This part of the text will be removed along with Figure 12.

[RC]: P18, table 1: I don't get how the GOSAT a-priori profile can outperform the mmc-IFS@MIPAS for the variability and mmc-IFS@ACE-FTS for mean bias and variability and even reaches the same performance as the mmC-IFS case. What does that mean? Can you comment on that?
[AR]: The GOSAT prior data is based in some version of C-IFS for the tropospheric data and TOMCAT for the stratosphere which is in turn constrained by the ACE-FTS observations. So, we'd expect it to do well.

[RC]: Figure 5 can be removed. The information content is too low and is already described in the text.
[AR]: Figure 5 will be removed

[RC]: I would like to see figure 1,2 in the unit molec/cm$^2$ this would allow to relate the partial columns to the total column.
[AR]: Plotting molecules/cm$^2$ as a map would end up showing just elevation, this being especially true for the troposphere (See: http://www.atmos-chem-phys.net/5/941/2005/acp-5-941-2005.pdf). Using the molecules/cm$^2$ metric would make sense if the measurement were made on a single spot on the globe, e.g by an FTIR instrument and not if the surface pressure is variable as measured by a satellite. Hence, we chose to plot the mixing ratios in ppm instead. The stratosphere was plotted in a similar fashion for consistency.

[RC]: figure 4 must be improved. The most interesting latitude range 30 - 55 degree is covered by the other data and cannot be seen in this representation.
[AR]: Done. The figure will be replotted for specific latitude bands.

**Reviewer # 2**

[RC]: As the MIPAS data are used as 'the reference' to quantify bias and random errors of the (extended) stratospheric CH4 data, a more detailed about the quality of MIPAS data is helpful. Also, the errors of the

MIPAS data should be taken into account when the authors discuss whether the precision and accuracy of the resulting stratospheric $CH_4$ columns meet the requirement for satellite $XCH_4$ validations etc.

[AR]:

Part1: Text will be modified to incorporate this information about errors in MIPAS.

Part 2: One of the limitations of stratospheric completion, in general, is that there is a lack of unbiased measurements with sufficient spatial coverage which can be used as reference truth. The "climatology" part (Section 3.3) of the paper still shows that having a few very accurate, unbiased measurements (like from balloons or AirCore) are still not sufficient in many regions, due to the variability in time and space of the stratospheric

10 component. It was for this reason that MIPAS, that provides nearly global coverage, was taken to be the "truth" in this study. But given this limitation, it would be best to develop models consistent with balloon- or AirCore-like measurements, to bridge this scale gap.

[RC]: I'd like some discussions on using the extended aircraft data for validating $CH_4$ retrievals from other

15 instruments such as IASI, particularly over extra-tropical regions.

[AR]: The following text will be added to the discussion part of the manuscript

[ME]: "This approach might also be used to validate remote sensing measurements from thermal infrared (TIR) sensors such as IASI or AIRS. Compared to near-infrared sensors, TIR measurements have an averaging kernel peaking relatively high in the atmosphere, generally around 200 mb in the Tropics and 400 mb toward the poles,

20 which makes the question of stratospheric completion all the more critical, as a comparatively large part of the signal is above the height of aircraft profiles, particularly in the Tropics. This can be mitigated by using measurements from high-altitude aircraft, such as from the HIPPO campaigns (Wofsy et al., 2012), with profiles extending up to 14 km, higher than that of commercial passenger aircraft. Profiles of methane measurements from HIPPO flights have been used to validate $CH_4$ retrievals from the thermal infrared sounder IASI (Xiong et

25 al., 2013). In this case monthly means from an atmospheric general circulation model were used to extrapolate from the ceiling of the aircraft profile to the top of the atmosphere. García et al. (2017) also used HIPPO measurements to validate CH4 retrievals from IASI, but instead chose to base their stratospheric extension on monthly and zonally averaged climatologies based on ACE-FTS measurements. The current study suggests that a certain uncertainty, particularly due to unresolved variability, is introduced by either of these climatology-based

30 approaches. Restricting the comparisons to the extratropics, where the aircraft profile covers more of the atmospheric column to which TIR sensors are sensitive, would also help to minimize the error introduced by stratospheric extension."

[RC] Line 20, Page 5: '..were specified from monthly global mean observations.. Please specify which CH4 observations were used to construct the global monthly mean'

[AR]: Done

[ME]:" The tropospheric mixing ratios of long-lived source gases, including $CH_4$, $N_2O$ and halocarbons, were specified from monthly global mean observations of these tracers."

[RC] Line 25, Page 5:'..which has no additional constraint...' To be more accurate, please add 'in stratosphere'

[AR]: Done

[RC] Line 27, Page 7:'....stratospheric CH4 that is likely to be due to the impact of the polar vortex dynamics', Change to '...stratospheric CH4, which is likely to be due to the impact of the polar vortex dynamics'

[AR]: Done

[RC] Line 27, Page 9: '...therefore, any variation in the bias along the latitude will be smoothed out', Why averaging along longitude will 'smoothing out any variations along the latitude' ?

[AR]: The text will be modified as follows:

[revised manuscript text omitted]

~~It is noteworthy that correction of model bias prior to using the fields for completion of aircraft profiles is expected to further improve these error estimates. We test this by applying a bias correction to the C-IFS model output for the year 2010. Fig. 12(a) shows the zonally averaged stratospheric column bias in the C-IFS model after application of the bias correction. We see that the overall magnitude of the bias in the stratospheric column reduces from 15 ppb pre-bias correction to less than about 8 ppb post-bias correction. Figure 12(b) shows the distribution of the stratospheric column bias computed at the locations where the MOZAIC aircraft profiles are measured. We see that the bias reduces greatly to -3.7 ppb while the random error reduces slightly to 5.7 ppb when bias corrected model data is used.~~

**3.3 Climatology-based approaches**

We now explore the potential of climatology-based approaches as stratospheric extensions for the aircraft profiles that, for instance, could be based on balloon-based measurements, satellite limb soundings or those from AirCore. Climatology based measurements are typically long term averages having a much sparser global coverage compared to global model output. For this part of the study, no real observations are used and we only evaluate the contribution of sparse data coverage and temporal averaging to the stratospheric column uncertainty. In order to do this, we analyse two main cases:

1. mmC-IFS: In this case, we use monthly mean C-IFS fields for our stratospheric assumption instead of full C-IFS fields with 6-hourly output (the FULL C-IFS case). This means that we do not account for the synoptic scale variability in the CH$_4$ vertical distribution. This helps us examine the impact of temporal variability of the data source on the stratospheric column bias.

[revised manuscript text omitted]

The approach of stratospheric completion might also be used to validate remote sensing measurements from thermal infrared (TIR) sensors such as IASI or AIRS. Compared to near-infrared sensors, TIR measurements have an averaging kernel peaking relatively high in the atmosphere, generally around 200 mb in the Tropics and 400 mb toward the poles, which makes the question of stratospheric completion all the more critical, as a comparatively large part of the signal is above the height of aircraft profiles, particularly in the Tropics. This can be mitigated by using measurements from high-altitude aircraft, such as from the HIPPO campaigns (Wofsy et al., 2012), with profiles extending up to 14 km, higher than that of commercial passenger aircraft. Profiles of methane measurements from HIPPO flights have been used to validate $CH_4$ retrievals from the thermal infrared sounder IASI (Xiong et al., 2013). In this case monthly means from an atmospheric general circulation model were used to extrapolate from the ceiling of the aircraft profile to the top of the atmosphere. García et al. (2017) also used HIPPO measurements to validate CH4 retrievals from IASI, but instead chose to base their

stratospheric extension on monthly and zonally averaged climatologies based on ACE-FTS measurements. The current study suggests that a certain uncertainty, particularly due to unresolved variability, is introduced by either of these climatology-based approaches. Restricting the comparisons to the extratropics, where the aircraft profile covers more of the atmospheric column to which TIR sensors are sensitive, would also help to minimize the error introduced by stratospheric extension.

In the coming years, an increased number of aircraft profiles of greenhouse gases, for instance, those from the IAGOS project, are expected to be available. Besides having great potential for providing robust validation methodologies of remote sensing observations and atmospheric models, these measurements have applications in NWP (e.g. in bias correction schemes or for data assimilation) as explored by the CAMS system. This can go a long way in contributing to an integrated global observing system and providing deeper insights into the chemical and physical processes in the atmosphere.

**Acknowledgements**

We thank Wuhu Feng (Leeds) for help with the TOMCAT model, which was supported by NCAS.

We thank the KIT-IMK team for making the MIPAS methane data available to us.

The Atmospheric Chemistry Experiment (ACE), also known as SCISAT, is a Canadian-led mission mainly supported by the Canadian Space Agency and the Natural Sciences and Engineering Research Council of Canada. We thank the ACE-FTS science team for providing methane data for this study.

[revised manuscript text omitted]